# Cadmium Sulfide—Reinforced Double-Shell Microencapsulated Phase Change Materials for Advanced Thermal Energy Storage

**DOI:** 10.3390/polym15010106

**Published:** 2022-12-27

**Authors:** Shendao Zhang, Yucao Zhu, Huanzhi Zhang, Fen Xu, Lixian Sun, Yongpeng Xia, Xiangcheng Lin, Hongliang Peng, Lei Ma, Bin Li, Erhu Yan, Pengru Huang

**Affiliations:** 1School of Material Science & Engineering, Guilin University of Electronic Technology, Guilin 541004, China; 2Guangxi Key Laboratory of Information Materials, Guangxi Collaborative Innovation Center of Structure and Property for New Energy and Materials, Guilin University of Electronic Technology, Guilin 541004, China

**Keywords:** microcapsules, double-shell, cadmium sulfide, photothermal conversion, heat storage

## Abstract

Phase change materials (PCMs) are widely used to improve energy utilization efficiency due to their high energy storage capacity. In this study, double-shell microencapsulated PCMs were constructed to resolve the liquid leakage issue and low thermal conductivity of organic PCMs, which also possess high thermal stability and multifunctionality. We used assembly to construct an inorganic–organic double shell for microencapsulate PCMs, which possessed the unprecedented synergetic properties of a cadmium sulfide (CdS) shell and melamine–formaldehyde polymeric shell. The scanning electron microscopy (SEM) and transmission electron microscopy (TEM) images confirmed the well-designed double-shell structure of the microcapsules, and the CdS was successfully assembled as the second shell on the surface of the polymer shell. The differential scanning calorimeter (DSC) showed that the double-shell microcapsules had a high enthalpy of 114.58 J/g, which indicated almost no changes after experiencing 100 thermal cycles, indicating good thermal reliability. The microcapsules also showed good shape stability and antileakage performance, which displayed no shape change and leakage after heating at 60 °C for 30 min. In addition, the photothermal conversion efficiency of the double-shell microcapsules reached 91.3%. Thus, this study may promote the development of microencapsulated PCMs with multifunctionality, offering considerable application prospects in intelligent temperature management for smart textiles and wearable electronic devices in combination with their solar thermal energy conversion and storage performance.

## 1. Introduction

The fast development of societies has induced increasing energy demand, making limited fossil fuels unable to meet the needs of development. Therefore, the utilization of sustainable and renewable energy is needed to improve energy utilization efficiency [1,2]. Thermal energy storage technology can store and release heat when the phase change procedure occurs [3,4]. Therefore, numerous heat storage materials, such as phase change materials (PCMs), have been widely used due to their high energy storage density and good thermal and chemical stability, which can obviously reduce energy consumption due to the phase change process by maintaining small fluctuations between indoor and outdoor temperatures [5,6]. For example, *n*-octadecane has been regarded as one of the best phase change energy storage materials due to its high heat storage capacity of 237.50 J/g with a phase transition temperature of 29.0 °C as well as stable chemical and noncorrosive properties [7,8]. However, special encapsulation technology for PCMs requires the prevention of flowability and leakage problems of *n*-octadecane during the phase change procedure. As a result, microencapsulation has recently attracted significant attention as an effective method to prevent the leakage of PCMs, which have been commonly used to encapsulate solid–liquid transition PCMs.

Microencapsulation technology has received considerable attention due to the resulting core–shell structure and attractive properties, which can supply good protection for PCMs to solve the leakage problem, and can also provide PCMs with high thermal cycling stability, a relatively constant volume, and a large heat transfer area [9,10]. Hussain et al. [11] prepared microencapsulated PCMs with double functionality using SiO_2_/SnO_2_ as the shell material, which exhibited a homogeneous spherical structure and high overall performance even after 1000 thermal cycles. Huang et al. [12] prepared novel microencapsulated PCMs by adding 3.34 wt% carbon nanotubes, which showed a high latent heat value of 208.9 J/g. According to reported research, most microcapsules were composed of a polymer shell to prevent PCM leakage due to their good physical and chemical stability. However, the facile flammability, poor thermal stability, and low thermal conductivity of polymers will naturally limit the widespread use of PCMs. Furthermore, single organic shell materials have a difficult time fulfilling the high mechanical strength and multifunctionality requirements for wide applications [13]. Therefore, various shell materials for microencapsulated PCMs have been widely studied, such as polymers, inorganic materials, and their hybrids. Liu et al. [14] studied a novel type of microcapsule based on an n-eicosane core and a phenol–formaldehyde resin shell, which exhibited excellent thermal reliability even after 500 thermal cycles. Hu et al. [15] prepared microencapsulated PCMs derived from an *n*-octadecane core and CaF_2_ shell, which showed an encapsulation ratio of 54.02% and a latent heat of 131 J/g with negligible leakage.

Inorganic materials or metal materials possess excellent thermal and mechanical stability. As a result, they have been introduced into the shell layer of microencapsulated PCMs to improve the drawbacks of single-polymer shells [16,17]. Among the reported microencapsulated PCMs, inorganic nanomaterials have always been suggested in the doping of the shell material. Their additional amount has been quite minimal, which could not effectively improve the performance of the composite PCMs. Although some researchers have fabricated inorganic shells to encapsulate PCMs to enhance the overall thermal conductivity of the microcapsules, they did not have the satisfactory compactness and conformability of organic PCMs since most inorganic shells have been synthesized through the sol–gel method. Therefore, double layers of shell materials have motivated many researchers to fabricate microencapsulated PCMs with excellent thermal stability and multifunctionality, which has also helped to expand the application fields of PCMs. Zhang et al. [18] first fabricated microencapsulated PCMs with a SiO_2_ shell as the first shell and with silver forming the second shell. Zhang et al. [19] prepared a new microencapsulated phase change material that was fabricated by in situ polymerization using paraffin as the core and a hydrophobic-silicon carbide-modified melamine-formaldehyde (MF) resin as the shell material, which had excellent mechanical properties and 58% enhanced thermal conductivity. Gu et al. [20] created double-shell microcapsules (DSMs) with both temperature control and low infrared emissivity characteristics, using eicosane as the PCMs and melamine, urea, and formaldehyde as the shell materials to form microcapsules. Subsequently, polyaniline was deposited on the surfaces of these microcapsules to form DSMs after coating polyester fabrics with the DSMs, which gave the fabric an obvious infrared stealth effect. However, research on organic–inorganic double-layer shell microcapsules has been rarely reported due to weak interfacial adhesion between the organic phase and the inorganic phase in the fabrication process [21]. Therefore, organic–inorganic double-layer shell microencapsulated PCMs still require further study to improve the energy storage capacity.

Cadmium sulfide (CdS) offers broad application prospects in the fields of solar energy conversion and photocatalysis due to its unique visible light absorption properties [22,23,24,25]. As a type of inorganic material, CdS combined with PCMs can endow PCMs with good thermal stability and highly efficient photothermal conversion. Nevertheless, it has been seldom used as the shell material to encapsulate PCMs for solar energy storage. Therefore, in this study, we integrated the respective characteristics of organic and inorganic materials mentioned above and fabricated novel microencapsulated PCMs with an organic–inorganic double-layer shell structure using polymer and CdS nanoparticles, which could improve the thermal stability and energy storage capacity of microencapsulated PCMs and provide microcapsules with multifunctionality. This combination of polymer and CdS nanoparticles could generate a synergetic effect to accelerate an effective solar light-to-heat conversion, affecting the widespread applications of microencapsulated PCMs.

Herein, microencapsulated *n*-octadecane was designed with MF polymer as the first compact polymer shell with CdS nanoparticles as the promising optimal shell. The effects of different CdS nanoparticle contents on the morphology, chemical properties, thermal storage properties, and optical properties of the double-shell microcapsules were systematically investigated. The results showed that the CdS nanoparticles were deposited on the polymer shell and formed into a homogeneous shell, which supplied the microcapsules with high photothermal conversion efficiency, and the melting latent heat of the microencapsulated PCMs was in the range of 113.4~154.94 J/g.

## 2. Experiment

### 2.1. Materials

Melamine (99%), the sodium salt of poly (styrene sulfonic acid) (C_8_H_7_NaO_3_S, PSS, Mw = 70,000), sodium sulfide nonahydrate (Na_2_S·9H_2_O, 99%), and cadmium chloride (CdCl_2_, 99.99%) were all provided by Shanghai Aladdin Biochemical Technology Co., Ltd. (Shanghai, China). Triethanolamine (98%) and formaldehyde (37–40%) were supplied by XILONG Scientific Co., Ltd. (Shantou, China). The sodium salt of poly (styrene maleic acid) (SMA, 19%) was provided by the Shanghai Leather Products Chemical Plant.

### 2.2. Preparation of the CdS-Reinforced Double-Shell Microencapsulated PCMs

#### 2.2.1. Preparation of the MF Resin Microcapsules

First, 10 mL of formaldehyde and 6 g melamine were homogeneously mixed in 10 mL of deionized water under mechanical stirring, and the pH value was adjusted to 8–9 with triethanolamine. Then, the mixture was reacted at 70 °C until the solution was completely transparent, producing the prepolymer solution of the polymer shell. Meanwhile, 25 g of *n*-octadecane, 7.5 g of SMA, and 150 mL of deionized water were added to a three-necked flask under vigorous stirring at 70 °C for 3 h to form a stable emulsion. Subsequently, the prepared prepolymer was added to the emulsion, the pH value was adjusted to 5–6 with citric acid, and the stirring speed was decreased to 600 r/min. After reacting for 6 h, the pH value of the mixture solution was adjusted to 8–9 using triethanolamine to end the reaction, and the melamine–formaldehyde resin microcapsules were produced after cleaning and drying.

#### 2.2.2. Preparation of the CdS-Reinforced Double-Shell Microencapsulated PCMs

The MF polymeric microcapsules and PSS with a mass ratio of 5:1 were evenly dispersed in 0.5 M Na_2_S solution for surface treatment to endow the microcapsules with negatively charged functional groups on the surface. Then, the modified microcapsules were homogeneously dispersed in deionized water, forming a stable dispersion. Next, 0.75 g/mol of cadmium chloride (CdCl_2_) and sodium sulfide (molar mass ratio of 4:1) were added dropwise to the microcapsule suspension under slow stirring for 2 h, producing the second CdS shell on the surface of the microcapsules. The theoretical mass fractions of the CdS shell in the microcapsules were 0 wt%, 5 wt%, 10 wt%, 15 wt%, 18 wt%, and 20 wt%.

### 2.3. Characterization

X-ray diffraction (XRD, Bruker-D8 Advance) and Fourier transform infrared spectroscopy (FT-IR, Nicolet 6700) were used to characterize the crystal structure and chemical composition. Scanning electron microscopy (SEM, FEI, Quanta FEG 450) was used to observe the micromorphology of the PCMs under 20 kV accelerating voltage. A differential scanning calorimeter (DSC TA Q250) was used to study the thermal properties of the samples, including the heat storage enthalpy, phase transition temperature, and thermal cycling stability in a temperature range of −10 °C to 120 °C at a heating rate of 10 °C min^−1^ under an N_2_ atmosphere. The samples were sputtered with gold–palladium alloy before testing. The microstructures of the PCMs were detected by transmission electron microscopy (TEM) (Hitachi JEM-1200EX, JEOL Ltd., Tokyo, Japan) operating at an accelerating voltage of 120 kV. An ultraviolet–visible–near–infrared (UV–Vis–NIR) spectrophotometer was used to measure the light absorption properties of the samples according to the diffuse reflection method. In addition, the solar thermal conversion performance was tested using a custom-built platform, equipped with a xenon light source as simulated sunlight with a wavelength of 200–600 nm. Subsequently, a 2 g sample was tested with an optical density of 100 mW/cm^2^, and the temperature–time curves were collected using a multichannel digital acquisition instrument. In addition, a high- and low-temperature testing box was used to simulate the actual application environment with a multichannel display instrument, which recorded and collected the tested data to characterize the temperature control and thermal management performance of the PCMs. An SDT Q600 thermogravimetric analyzer was used to test the thermal stability of the samples to 700 °C at a heating rate of 10 °C/min under nitrogen protection.

## 3. Results and Discussion

### 3.1. Schematic Fabrication of the CdS-Reinforced Double-Shell Microencapsulated PCMs

The fabrication of CdS-reinforced double-shell microencapsulated PCMs is illustrated in Figure 1. First, the polymeric microcapsules were prepared through in situ polymerization using MF resin, as reported in the references [25]. The unique and hydrophobic groups of SMA were used as cross-linkers to wrap the *n*-octadecane oil phase inside the stable micellar solution, and the hydrophilic groups were assembled with the MF prepolymer through interfacial reactions among the functional groups. As a result, the compact melamine–formaldehyde polymeric shell was formed to prevent the leakage problem of *n*-octadecane and improve the thermal stability of the microencapsulated PCMs. Subsequently, PSS was used as a surface treatment agent, which could form a surfactant layer on the surface of microcapsules to induce the self-assembly of the CdS second shell. Therefore, sulfonate groups were formed on the surface of the microcapsules after suspension in PSS aqueous solution, and the MF polymeric microencapsulated PCMs were positively charged. As a result, Cd^2+^ adsorbed on the surfaces of the microcapsules through complexation and attraction of the positive and negative charges. Finally, Cd^2+^ on the surfaces of the microcapsules was reduced to CdS via the redox reaction as sodium sulfide was added so that the formed nano CdS was gradually deposited on the surfaces of the polymer microcapsules would form the CdS-reinforced double-shell microcapsules. The compact polymeric shell maintained the thermal stability of the microcapsules, and the second CdS shell endowed the microcapsules with good photothermal conversion and comprehensive performance.

### 3.2. Microstructures of the CdS-Reinforced Double-Shell Microencapsulated PCMs

Figure 2 shows the FT-IR spectra of the CdS, *n*-octadecane, and double-shell microcapsules. We observed that the C–H stretching vibrations and in-plane and out-plane bending vibration peaks of *n*-octadecane appeared at 2923 cm^−1^, 2853 cm^−1^, 1469 cm^−1^, 1341 cm^−1^, and 719 cm^−1^ while a weak absorption peak of CdS presented at 618 cm^−1^, corresponding to the stretching vibration of CdS. The FT-IR spectra of the microcapsules without nano-CdS showed that the bending vibration peaks of aliphatic C-N appeared at 1159 cm^−1^ and 1014 cm^−1^, the stretching vibration absorption peaks of O-H and N-H appeared at 3413 cm^−1^, and the multiple stretching vibration peaks of C-N appeared at 1555 cm^−1^, indicating that melamine and formaldehyde successfully reacted [19,26], forming the polymeric shell material through in situ polymerization. Furthermore, all the characteristic peaks of *n*-octadecane could be found in the FT-IR spectrum of the microcapsules. These results inferred that the MF polymer-based microcapsules were successfully synthesized. In addition, compared with the double-shell microcapsules with different CdS shell material contents, we found the absorption peaks of both the *n*-octadecane and MF resins, and a weak absorption peak of CdS was also observed at 618 cm^−1^, indicating that the CdS had been successfully integrated into the microcapsules. These results confirmed that the CdS-enhanced double-shell microcapsules were fabricated [27].

The chemical composition elements of the double-shell microcapsules with 18 wt% CdS were characterized by XPS. According to the split peak spectrum in Figure 3a, the characteristic peaks of the Cd 3d_5/2_ and Cd 3d_3/2_ orbitals appeared at binding energies of 404.7 eV and 411.5 eV, respectively. At the same time, two deconvoluted signals of the S_2p_ component appeared at 161.3 eV and 162.6 eV, as clearly shown in Figure 3b, indicating the presence of CdS [28]. As shown in Figure 3c, there were three chemical states of C at 287.7 eV, 286.2 eV, and 284.9 eV, corresponding to the characteristic peaks of O–C=C, C–O–C, and C–C, respectively. In addition, C–O with a binding energy of 532.4 eV was observed, as shown in Figure 3d. These results indicate that CdS was successfully added to the microcapsules, indicating the successful fabrication of the CdS-reinforced microencapsulated PCMs with a double-shell material.

The micromorphology and microstructure of the CdS-reinforced double-shell microcapsules were characterized by SEM and TEM. The SEM images in Figure 4 show that all of the prepared samples had a spherical core–shell structure, and their particle size was uniformly distributed in a concentration range of about 3–10 μm, which could effectively encapsulate *n*-octadecane. It was evident that some prepolymer particles adhered to the surfaces of the microcapsules without CdS. In addition, the surfaces of the microcapsules became relatively dense and smooth with the increasing addition of CdS, and the spherical structures became more regular. However, the agglomeration phenomenon among the microcapsules became serious when the addition content of CdS increased up to 20 wt% due to the agglomeration effect of the CdS nanoparticles, where the microcapsule surfaces could not provide sufficient electrochemical sites for CdS. Therefore, 18 wt% CdS was a suitable addition amount for the preparation of the double-shell microcapsules.

TEM was further used to confirm the core–shell structure of the double-shell microcapsules, as shown in Figure 5. It was clear that the microcapsules presented a standard spherical core–shell structure. In comparison, the microcapsules with only the MF polymeric shell possessed a thin grey shell, as shown in Figure 5b. We found that the thickness of the shell material for the CdS-reinforced double-shell microcapsules was thicker than that of the microcapsules without CdS. Furthermore, it was important that the compactness of the shell material significantly improved. As calculated, the shell thickness of the double-shell microcapsules with 18 wt% CdS was 154.194 nm, which provided sufficient mechanical strength and structural stability for the microencapsulated PCMs.

To estimate the distribution homogeneity of CdS in the microcapsules, the elemental distribution of the double-shell microcapsules with 18 wt% CdS was further tested by EDX, as shown in Figure 5. Notably, the microcapsule shell was composed of Cd, S, C, and O elements, and Cd and S were uniformly distributed over the entire microcapsule surface. This result further indicates that CdS was deposited on the surfaces of the polymer microcapsules in a regular and orderly manner through charge interactions to form the second shell of the microcapsules, laying the foundation for the high performance and functionality of the microcapsules.

Figure 6 shows the XRD patterns of *n*-octadecane, the double-shell microencapsulated PCMs, and CdS. Distinctive characteristic peaks were observed for *n*-octadecane at 2θ of 19.31°, 19.67°, 23.46°, and 24.53°, which were generated by the crystal transition of *n*-octadecane during homogeneous nucleation and corresponded to the (011), (012), (101), and (102) crystal planes, respectively. These diffraction peaks indicate that the crystal structure of *n*-octadecane did not change, as confined in the double shell of the microcapsules, and the crystalline properties were well maintained. Meanwhile, the characteristic peaks corresponding to the (111), (220), and (311) crystallographic planes of CdS also appeared at 2θ of 26.57°, 44.09°, and 52.81° in the diffraction patterns of the double-shell microcapsules, indicating the cubic phase structure of the CdS shell [28]. These results further indicate that CdS was successfully assembled on the surface of the polymer shell forming the second shell of the microcapsules, ensuring the microcapsules with good thermal performance and excellent photothermal properties.

### 3.3. Thermal Properties of the CdS-Reinforced Double-Shell Microencapsulated PCMs

Heat storage capacity and phase change temperature are the most vital performance metrics for PCMs, playing a vital role in determining their practical applications. Therefore, DSC was performed to evaluate the endothermic/exothermic processes and energy storage values of the CdS-reinforced double-shell microencapsulated PCMs, as shown in Figure 7. We observed that a single peak appeared in the endothermic and exothermic processes of *n*-octadecane, according to its good phase transition process. Three exothermic peaks appeared for the microencapsulated samples during the crystal process [29] where the α-peak and β-peak corresponded to the heterogeneous nucleation process of *n*-octadecane due to confinement in the shell material of the microcapsules. In addition, the γ peak was attributed to the phase transition of *n*-octadecane from melting to the crystalline state because of homogeneous nucleation. Compared with the polymeric microcapsules without CdS, the double-shell microcapsules showed similar characteristic peaks with the polymeric microcapsules, indicating their good phase change behavior. This also confirmed that the CdS shell did not decrease the phase transition behavior of the microcapsules, which possibly enhanced the thermal stability of the microcapsules.

Table 1 shows the latent heat and phase change temperatures of the microencapsulated PCMs during the fusion and crystallization processes. It was clear that the crystallization and fusion enthalpy of the CdS-reinforced microcapsules showed a small decrease with increasing CdS content due to the reduction in weight percent of the PCMs in the microcapsules. Nevertheless, the experimental latent heat of the double-shell microcapsules was essentially the same as the evaluated value according to the PCM weight percent. Significantly, the polymeric microcapsules possessed melting and crystal latent heat values of 169.06 J/g and 168.55 J/g. When the CdS content was 5 wt%, the enthalpy values of the double-shell microcapsules were 153.23 J/g and 154.94 J/g, and when the content of CdS increased to 10 wt%, the enthalpy values improved to 141.63 J/g and 143.13 J/g. When the CdS content was 15 wt%, the enthalpy values were 129.26 J/g and 129.12 J/g. Of note, the enthalpy values of the dual-shell microcapsules were 114.17 J/g and 114.58 J/g due to the increased CdS shell content of 18 wt%. As the CdS shell content further increased up to 20 wt%, the microcapsules showed little change in enthalpy compared with the microcapsules with 18 wt% CdS because some of the CdS nanoparticles adhered on the surfaces of the microcapsules, which did not form into the shell material. Therefore, 18 wt% CdS was the proper addition amount to form the second shell for the microencapsulated PCMs.

Furthermore, as compared with the melting temperature and the crystal temperature of the α-peak, we found that *n*-octadecane showed a supercooling degree of 5.99 °C. In addition, the polymeric microcapsules displayed a supercooling degree of 6.04 °C, which indicates that the confinement of the polymeric microcapsules had little effect on the phase change behavior of the interior *n*-octadecane. Moreover, the supercooling degree of the microencapsulated PCMs with different CdS shell contents showed significant changes. The supercooling degree of the microcapsules with 5 wt% decreased to 5.41 °C due to the good thermal conductivity of the thin CdS shell. As the CdS content increased to 10 wt%, 15 wt%, and 18 wt%, the supercooling degree of the microcapsules increased to 6.7 °C, 7.89 °C, and 6.69 °C, respectively, implying that increasing the CdS shell content reduced the thermal response rate due to its heat resistance. As the CdS content increased to 20 wt%, the supercooling degree of the microcapsules decreased to 5.76 °C because the excessive CdS cumulated and did not form into the thick shell material for the microcapsules. Therefore, 18 wt% CdS was the maximum additional amount to fabricate the microencapsulated PCMs with a good double-shell structure.

We also conducted 100 DSC thermal cycles of the microencapsulated PCMs with 18 wt% CdS shell content to evaluate the thermal cycling stability for practical applications, as shown in Figure 8. After 100 thermal cycles, there was almost no change in the melting curves, which confirmed that the CdS-reinforced double-shell microcapsules maintained good phase change behavior. In the cooling process, there was a remarkable decline in the α-peak intensity while the intensity of the γ-peak increased after 100 thermal cycles. This result indicates that the phase transition behavior of *n*-octadecane changed from homogeneous to heterogeneous nucleation after thermal cycling due to the induction of the double shell material of the microcapsules. Moreover, the γ-peaks and total area of all the peaks were almost unchanged, which verified that the phase transition behavior of the CdS-reinforced double-shell microcapsules was well maintained after thermal cycling, and the enthalpy retention rate was above 98.2% after thermal cycling. These results confirmed the excellent thermal cycling stability of the CdS-reinforced double-shell microcapsules, offering potential applications for thermal energy storage. The microstructures of the CdS-reinforced microcapsules were further tested using the IR spectra before and after thermal cycling. Notably, the intensity and position of the characteristic absorption peaks of the microcapsules did not change after 100 thermal cycles, indicating that no chemical reaction occurred, and no structural changes occurred during the cycling process. These results further proved that the CdS-reinforced microcapsules had excellent thermal cycle stability.

Figure 9 shows the TG curves for the *n*-octadecane and microcapsules with different CdS shell contents. It was evident that *n*-octadecane exhibited only one thermal degradation stage according to the evaporation and thermal degradation results of *n*-octadecane. The microcapsules were thermally decomposed in two stages, and we inferred that the first stage of thermal decomposition for the microcapsule sample was identical to the temperature range of *n*-octadecane. However, the initial degradation temperature was slightly lower than that of *n*-octadecane, indicating that the introduction of the CdS shell increased the thermal response rate of the microcapsules, which was helpful for the PCMs to undergo phase change immediately with a change in environmental temperature. The second weight-loss stage was caused by the decomposition of the melamine–formaldehyde resin shell of the microcapsules, which occurred in the temperature range of 210–500 °C. In addition, the remaining amount of *n*-octadecane increased with increasing CdS content. These results confirmed that CdS was successfully added as the second shell to the microencapsulated PCMs.

The temperature–time curves obtained by testing the double-shell microcapsules with 18 wt% CdS shell content and the polymeric microcapsules without CdS are shown in Figure 10. During the heating and cooling processes, we found that the temperature of the double-shell microcapsules with 18 wt% CdS shell content increased faster than the polymeric microcapsules, indicating that the CdS shell made the microcapsules more sensitive to temperature changes. Furthermore, we observed that the microcapsules started to show a similar temperature platform when the temperature increased to 20 °C, which was maintained in this temperature range for about 256 s and 249 s, respectively, for the CdS-reinforced microcapsules and the polymeric microcapsules. This was because the *n*-octadecane underwent phase change from a crystalline state to the melting state by absorbing heat from the environment, and this indicates that the CdS-reinforced double-shell microcapsules had temperature control properties as good as the polymeric microcapsules. During the cooling process, the durations of the temperature plateau were 126 s and 119 s at 20–30 °C and 97 s and 90 s at 10–15 °C for the CdS-reinforced microcapsules and polymeric microcapsules, respectively, which corresponded to the α and γ peaks of *n*-octadecane during crystallization. Furthermore, the temperature of the 18 wt% CdS double-shell microcapsules dropped faster than the polymeric microcapsules, which also demonstrated that the CdS shell material improved the thermal response sensitivity of the fabricated microcapsules.

### 3.4. Photothermal Conversion Performance of the Microcapsules

Because CdS possessed optical materials, it was an effective way to improve energy efficiency, which could supply the PCMs with a high solar thermal conversion efficiency. Figure 11a shows the diffuse light absorption diagrams of the CdS nanoparticles and the double-shell microcapsules with different CdS shell contents. The CdS nanoparticles possessed more than 75% light absorption in the light wavelength range of 200–500 nm, and the light absorption increased as high as 96% in the visible wavelength range of 400–500 nm, confirming the excellent light absorption properties of CdS. In addition, the ultraviolet (UV) light absorption rate of the CdS-reinforced double-shell microcapsules increased with increasing CdS shell content when the light wavelength was 200–400 nm. When the CdS content was 20 wt%, the UV absorption rate reached 53%. Moreover, the UV light absorption rate of the CdS-reinforced microcapsules was higher than 73% when the light wavelength was 400–500 nm, and the highest absorption rate was 85% in visible light. The abovementioned results indicate that the CdS shell provided the microcapsules with excellent light absorption characteristics. Figure 11b shows the temperature–time curves obtained by irradiating the samples under simulated sunlight with a light intensity of 100 mW/cm^2^ for 20 min. We significantly observed that the temperature of CdS increased rapidly under irradiation until the temperature reached approximately 60 °C. Similarly, the temperature of the polymeric microcapsules without the CdS shell also increased rapidly under irradiation for about 20 min; however, the final temperature only reached 37 °C. This result indicates that CdS could effectively capture photons, increasing its temperature and storing the absorbed energy. Furthermore, the light–thermal conversion and heat storage efficiency was calculated according to Equation (1):(1)$$\theta =\frac{m\cdot \Delta H}{P\;s\;\left(t_{t}-t_{f}\right)}\times 100\%$$
where *m* and Δ*H* are the quantity (2 g) and thermal energy storage enthalpy of the samples, respectively, *p* and *s* denote the light density (100 mW/cm^2^) and sample irradiation area (12.56 cm^2^), respectively, and *t_f_* and *t_t_* signify the initial and end times of the double-shell microencapsulated PCMs to complete phase transition under light irradiation, respectively. The calculated photothermal conversion efficiency rates of the double-shell microcapsules with CdS contents of 5 wt%, 10 wt%, 15 wt%, 18 wt%, and 20 wt% were 64.8%, 69.3%, 82.5%, 85.2%, and 91.3%, respectively. These results showed that the CdS-reinforced microcapsules had a high light-to-heat conversion and thermal storage efficiency. When the light source was turned off, the temperature of the CdS showed a sharp decrease, reaching room temperature quickly, while the temperatures of the microcapsule samples showed a slow decrease, which was slower with decreasing CdS shell content, and the temperature reached equilibrium at about 35 °C, indicating that the PCMs released stored energy to maintain their own temperature equilibrium at this time.

In addition, we compared the thermal performance results of the CdS-reinforced microencapsulated PCMs with those reported in the references, as shown in Table 2. It was clear that the light-to-heat conversion efficiency of the microencapsulated PCMs was higher than most of the references, indicating that the introduction of the CdS shell enhanced the light-to-heat conversion performance of the microcapsules. Furthermore, the thermal energy storage capacity of the microcapsules was approximately in line with the capacity shown in the reported results. This comparison further confirmed the advanced performance of the CdS-reinforced microencapsulated PCMs.

### 3.5. Shape Stability and Antileakage Properties of the Samples

To investigate the shape stability and antileakage properties of *n*-octadecane, the polymeric microcapsules, and the CdS-reinforced double-shell microcapsules for long-term practical applications, a digital camera was used to visualize the samples under heating at 60 °C. We clearly observed from Figure 12 that the majority of the *n*-octadecane melted into liquid after heating for 20 min. The microcapsules without CdS showed slight leakage after heating for 30 min, but the CdS-reinforced double-shell microcapsules maintained their initial solid shape well without liquid leakage after heating for 30 min. These results demonstrated the excellent shape stability of the CdS-enhanced double-shell microcapsules due to their well-constructed double-shell structure.

## 4. Conclusions and Future Perspectives

This study constructed CdS-reinforced double-shell microencapsulated PCMs by taking full advantage of organic and inorganic hybrid materials. In addition, CdS was successfully fabricated on the polymeric shell of the microcapsules, producing the second shell, which endowed the microcapsules with good shape and thermal stability and excellent solar thermal conversion performance. SEM and TEM images confirmed the well-designed double-shell structure of the microcapsules, and the CdS shell was successfully assembled as the second shell. DSC showed that the double-shell microcapsules had a high enthalpy value of 114.58 J/g, which showed almost no change after experiencing 100 thermal cycles. The microcapsules also show good shape stability and antileakage performance, which displayed no shape change and leakage after heating at 60 °C for 30 min. In addition, the photothermal conversion efficiency of the double-shell microcapsules reached 91.3%. Thus, this study may promote the development of microencapsulated PCMs with multifunctionality, giving them considerable application prospects in solar energy systems.

## Figures and Tables

**Figure 1 polymers-15-00106-f001:**
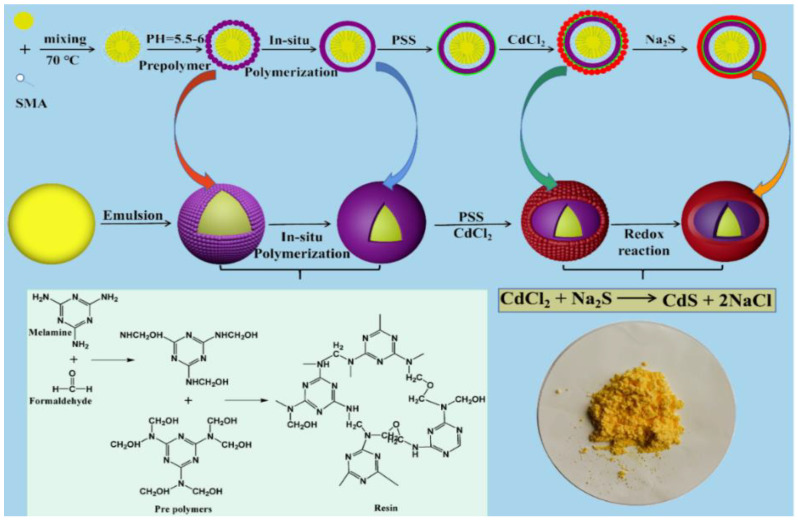
Synthetic mechanism of CdS double-shell microcapsule composite phase change material.

**Figure 2 polymers-15-00106-f002:**
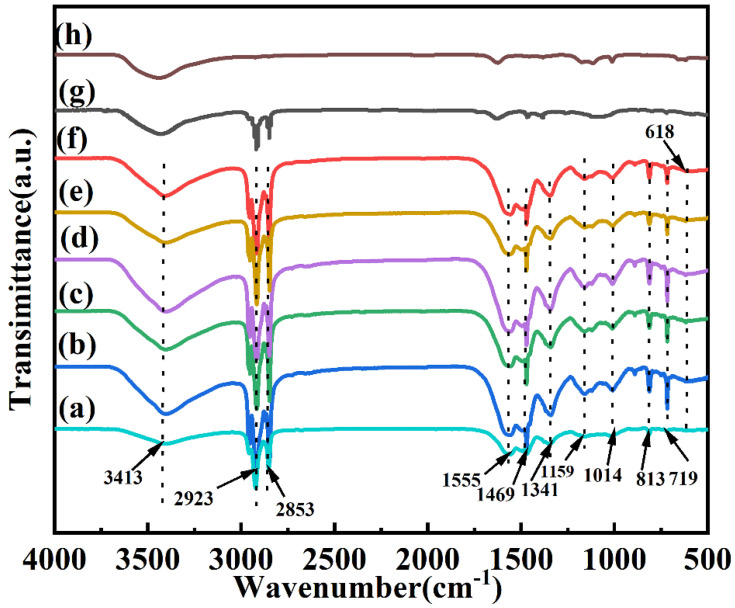
FT-IR spectra of the double-shell microencapsulated PCMs with different CdS contents: (**a**–**f**) 0 wt%, 5 wt%, 10 wt%, 15 wt%, 18 wt%, and 20 wt%, (**g**) *n*-octadecane, and (**h**) CdS.

**Figure 3 polymers-15-00106-f003:**
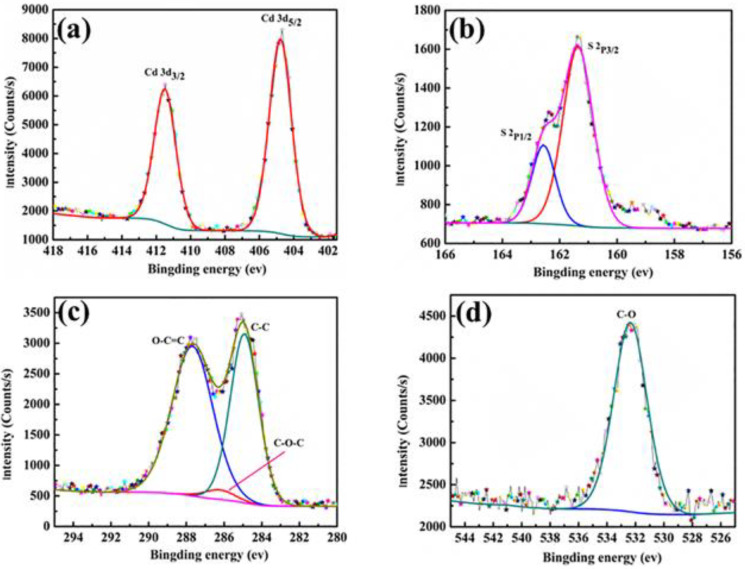
XPS spectra of double shell microcapsules with 18 wt% CdS: (**a**) Cd 3d; (**b**) S 2p; (**c**) C 1s; (**d**) O 1s.

**Figure 4 polymers-15-00106-f004:**
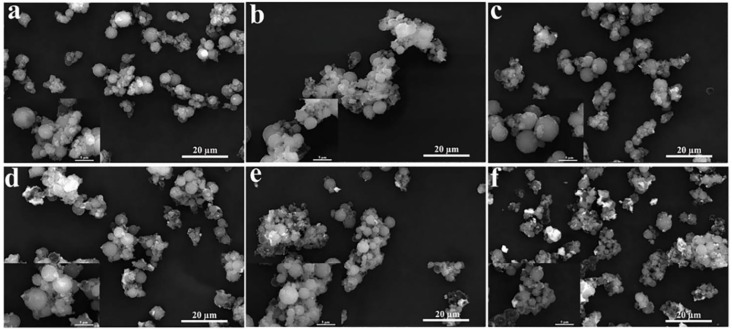
SEM images of double-shell microencapsulated PCMs with different CdS shell contents: (**a**–**f**) 0wt%, 5 wt%, 10 wt%, 15 wt%, 18 wt%, and 20 wt%.

**Figure 5 polymers-15-00106-f005:**
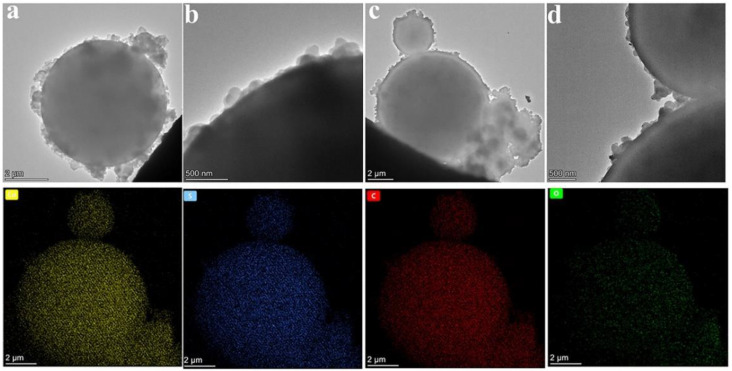
TEM micrographs of microcapsules: (**a**,**b**) polymeric microcapsules; (**c**,**d**) the double-shell microcapsules with 18 wt% CdS; and EDX images.

**Figure 6 polymers-15-00106-f006:**
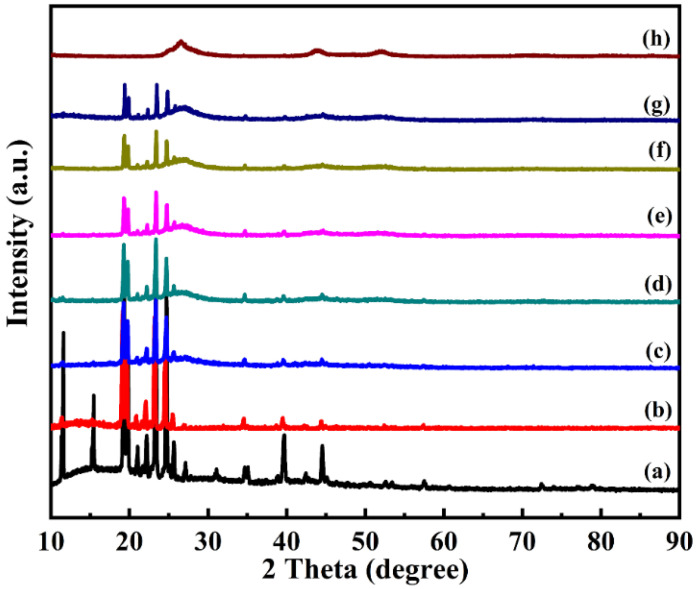
XRD patterns of the (**a**) *n*-octadecane and double-shell microencapsulated PCMs with different CdS shell contents of (**b**–**g**) 0 wt%, 5 wt%, 10 wt%, 15 wt%, 18 wt%, and 20 wt%, and (**h**) CdS.

**Figure 7 polymers-15-00106-f007:**
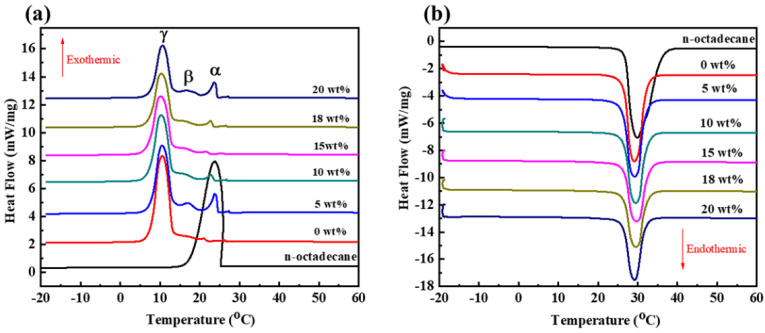
DSC thermograms of the *n*-octadecane and double-shell microencapsulated PCMs with different CdS shell contents: (**a**) cooling process and (**b**) heating process.

**Figure 8 polymers-15-00106-f008:**
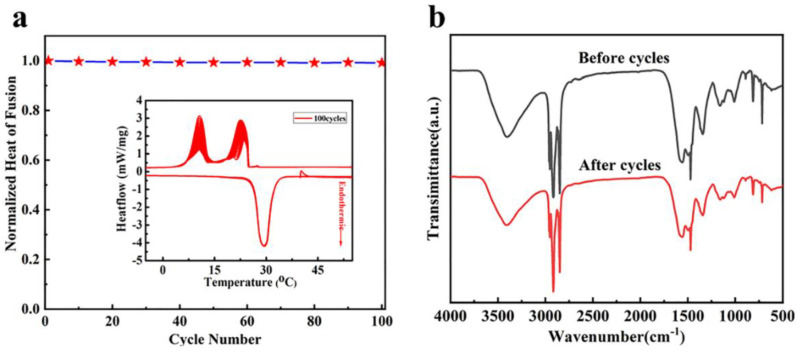
(**a**) 100 DSC thermograms of the double-shell microencapsulated PCMs with 18 wt% CdS shell content; (**b**) FT-IR spectra before and after 100 thermal cycles.

**Figure 9 polymers-15-00106-f009:**
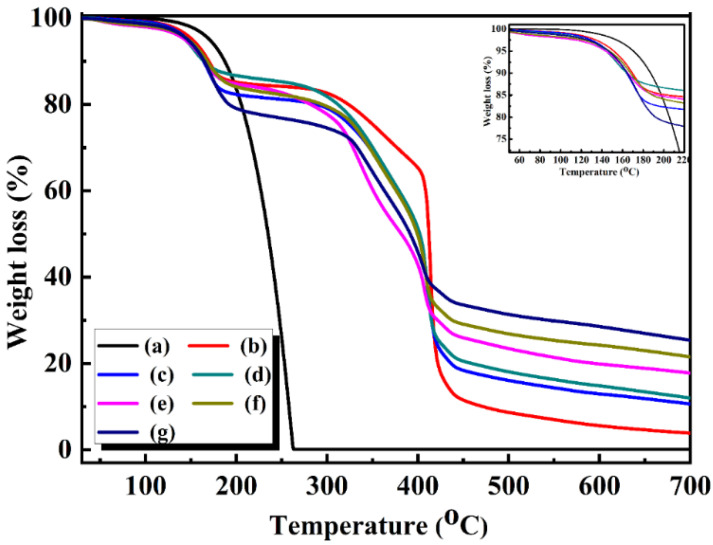
TG curves of *n*-octadecane and the double-shell microencapsulated PCMs with different CdS shell contents: (**a**) *n*-octadecane; (**b**) 0 wt%; (**c**) 5 wt%; (**d**) 10 wt%; (**e**) 15 wt%; (**f**) 18 wt%; (**g**) 20 wt%.

**Figure 10 polymers-15-00106-f010:**
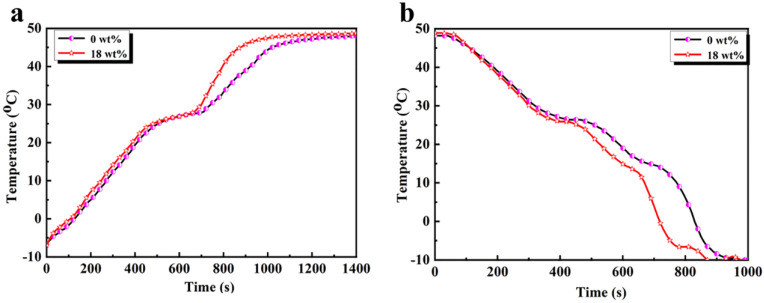
Temperature control curves of the double-shell microencapsulated PCMs with 18 wt% CdS shell content and the polymeric microcapsules without CdS: (**a**) heating process; (**b**) cooling process.

**Figure 11 polymers-15-00106-f011:**
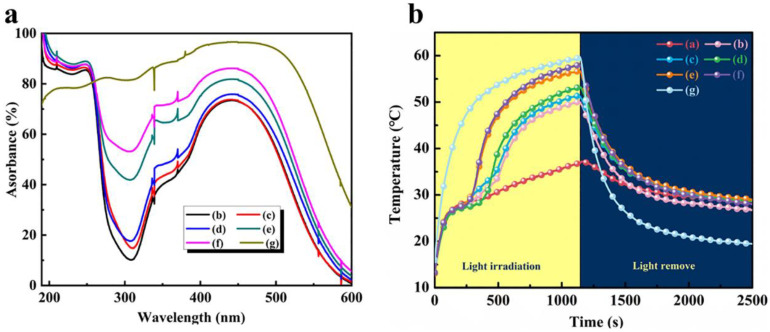
Light absorption curve (**a**) and temperature-time curve (**b**) of double-shell microencapsulated PCMs with different CdS shell contents under irradiation: (a) 0 wt%; (b) 5 wt%; (c) 10 wt%; (d) 15 wt%; (e) 18 wt%; (f) 20 wt%; (g) CdS.

**Figure 12 polymers-15-00106-f012:**
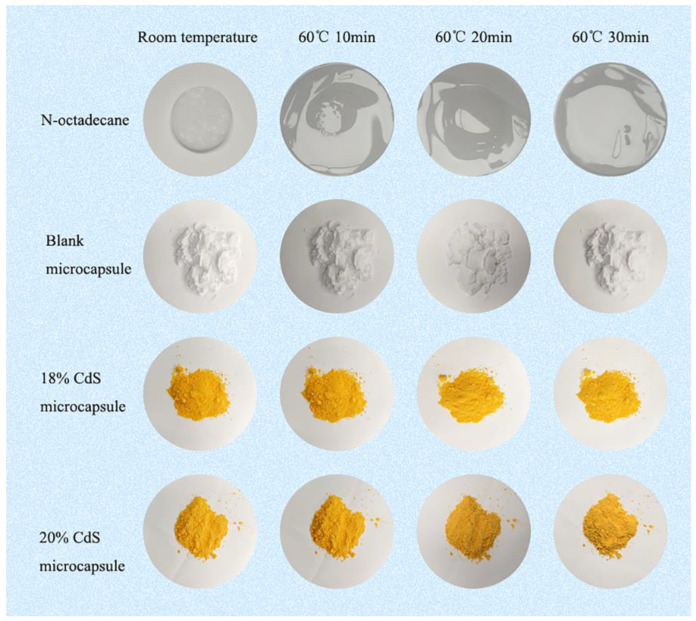
Shape stability of *n*-octadecane, polymeric microcapsules, and the CdS-reinforced double-shell microencapsulated PCMs.

**Table 1 polymers-15-00106-t001:** Phase transition properties of *n*-octadecane and CdS reinforced double-shell microencapsulated PCMs.

Sample Code	T_m_ (°C)	ΔH_m_ (J/g)	ΔH_m t_ (J/g)	Tc (°C)			ΔH_c_ (J/g)	ΔH_c.t_ (J/g)
				α	β	γ		
a	29.9	236.40	236.40	23.91	-	-	235.4	235.4
b	29.14	169.06	169.35	23.10	15.20	10.56	168.55	168.35
c	29.17	154.94	154.88	23.76	16.97	10.53	153.23	153.76
d	29.49	143.13	143.04	22.79	15.86	10.24	141.63	142.44
e	29.73	129.26	129.23	21.84	15.85	10.15	129.12	128.69
f	29.48	114.58	114.61	22.79	15.89	10.28	114.17	114.07
g	29.20	113.73	113.80	23.44	16.74	10.61	113.40	113.32

**Table 2 polymers-15-00106-t002:** Photothermal conversion of double shell microcapsules.

Microcapsules	Light-to-Thermal Conversion Efficiency	Latent Heat	Reference
N-eicosane@TiO_2_/TiN	78.4%	163 J/g	[30]
BN/SIO_2_@ *n*-octadecane	69.54%	140.6 J/g	[31]
MF/CuS@ dodecanol tetradecyl ester	85.60%	180.3 J/g	[32]
SiO_2_/Ti_4_O_7_@PA	85.36%	169 J/g	[33]
PU@butyl stearate containing	50.7%	72.14 J/g	[34]
nano-SiC/polystyrene@Octadecane	54.91%	105.7–106.3 J/g	[35]
nano-SiC/MUF@CA	74.40%	97.80 J/g	[36]
CdS/MF/*n*-octadecane	85.2%	114.58 J/g	This work

## Data Availability

Data is contained within the article.

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
