# Peer review of "Cadmium Sulfide—Reinforced Double-Shell Microencapsulated Phase Change Materials for Advanced Thermal Energy Storage"

_polymers, 2022, doi:10.3390/polym15010106_

Round 1

Reviewer 1 Report

1. The abstract doesn't speak about the specific outcomes. Rewrite the abstract with the quantitative results.

2. The whole need to be critically checked and corrected for the grammar and typos. For example, in introduction, 3rd paragraph, 22nd line, 'which has the fabric has an..' seems to be odd to understand. 

3. Mention the specific application for which the proposed PCM is studied. Since the requirement of PCM properties such as melting point, latent heat, chemical and thermal properties should vary for different solar energy applications. 

4. In 2.2.1, mention duration as 'hours' instead of 'h'

5. Why nanoparticle weight fractions were chosen as '0 wt%, 5 wt%, 10 wt%, 15 wt%, 18 wt% 

and 20 wt%'? Any specific reason?

6. In 3.1, it is claimed '18 wt% CdS is a suitable addition amount for the preparation of doubleshell microcapsules'. But no strong discussion for this claim is given. Further, from the SEM images, it seems that agglomeration is not alone caused at 20% it is also found in lower fractions too. Please give a clear justification.

7. Include the table showing physical and thermal properties (including melting point, solidifying point, thermal conductivity and latent heat) of raw ingredients, since they are not given anywhere in the text.

8. In the last line of introduction, it is mentioned 'microencapsulated PCMs was improved from 113.4 J/g to 154.94 J/g'. This statement looks contrast with the table 1. Justify.

9. The caption of Figure 7 looks confusing, since it is having two images a and b. It can be suitably modified.

10. From Fig. 7a, it seems that the inclusion of Cds significantly reduced the melting point and it seems near by 10 degree Celcius. If so, how is it suitable for a specific application? Please explain.

11. How theoretical latent heat was calculated? the comparison is given in table 1.

12. In fig. 8a, why there is two peaks in exothermic process? How?

13. Include a discussion of on a degree of supercooling, since the micro encapsulation seem to be caused the significant change in melting point.

14. Why uv light absorption is measured here, since the work is all about charging and discharging if the PCM and it may not have exposed to direct sunlight in most of the cases? Give the significance in detail.

15. In page 14, in equation 1, 'm' is mentioned as quality. Correct it.

16. Fig. 12 is no where explained in the text. Discuss on it.

17. How enthalpy value of 114.58 J/g is claimed to be high? Table 2 indicates ref. 35 and 36 have shown a very good efficiency as well as a very high latent heat. If so, What is the significance of this present work?

18. The conclusion doesn't convey the actual outcome of the work. Need to be re-written. 

Author Response

Dear Editor and reviewers:

    Thank you very much for your decision and constructive comments on our manuscript. We have carefully considered the suggestions and have made some corrections of the manuscript. We had tried our best to improve and made some changes in the manuscript.

       And we have revised this manuscript accordingly based on all the points that the reviewers suggested, and some complementary data have been presented in the revised manuscript. Thanks a lot for giving us one more opportunity to address the comments from the reviewers. We sincerely appreciate the reviewers for their constructive suggestions and hope that the revised manuscript is now acceptable.

You will find a highlighted version of our manuscript, which identify all the modification we made to the manuscript. If there are any other questions or further revision needed, please inform me in your early convenience. All your helps will be greatly appreciated. 

Reviewer 2 Report

Greetings, Editor thank you for providing me with the opportunity to re-review the article. I reviewed the article with ID= polymers-2083764. Overall, the article structure and content are suitable for the Polymers journal. Please consider these suggestions as listed below. I am sending again major revision to authors.

  1. The title seems ok but remove abbreviation from title.
  2. The abstract seems to be good. Just add an introductory line in the start.
  3. What is Research gap? What is need of this study? It is unclear still.
  4. Rewrite the keywords. Its seems very long. Maximum two words should be a keyword.
  5. Please highlight the novelty clearly in the introduction. Its seems there is a confusion.
  6. Please check the abbreviations of words throughout the article. All should be consistent.
  7. Graphical abstract is amazing. It is perfect.
  8. Please do not use lumpy references such as 1-4 at page 1. Please delete all and simply cite this single article here- Yaqoob, A.A.; Ahmad, A.; Ibrahim, M.N.M.; Karri, R.R.; Rashid, M.; Ahamd, Z. Chapter 23—Synthesis of metal oxide–based nanocomposites for energy storage application. In Micro and Nano Technologies, Sustainable Nanotechnology for Environmental Remediation; Koduru, J.R., Karri, R.R., Mubarak, N.M., Bandala, E.R., Eds.; Elsevier: Amsterdam, The Netherlands, 2022; pp. 611–635.
  9. The first paragraph has 12 references. Its seems very weird. Please remove all and maximum put 3 or 4 reference in this paragraph.
  10. Page 3 first paragraph need a reference. Cite this one- Yaqoob, A.A.; Ibrahim, M.N.M.; Ahmad, A.; Khatoon, A.; Setapar, S.H.M. Polyaniline-Based Materials for Supercapacitors. Handb. Supercapacit. Mater. Synth. Charact. Appl. 2021, 113–130.
  11. What is CdS?
  12. The main objective of the work must be written on the more clear and more concise way at the end of introduction section. Its weird in the present state.
  13. Please provide space between number and units. Please revise your paper accordingly since some issue occurs on several spots in the paper.
  14. Page 4, the last paragraph seems very weird please concise it to make a sense.
  15. Unit of temperature is wrong in several place. Please revise your paper accordingly since some issue occurs on several spots in the paper.
  16. So far results are explained well but there is lack of discussion. Why there is no scientific discussion. Author should verify the results instead of simple explaining.
  17. Add a comparison to earlier studies and evidence that this is the first time we are reporting on this topic.
  18. Regarding the replications, authors confirmed that replications of experiment were carried out. Please give a statement regarding this concern.
  19. Section 4 should be renamed by Conclusion and Future perspectives. Conclusion section is missing some perspective related to the future research work, quantify main research findings, highlight relevance of the work with respect to the field aspect.
  20. To avoid grammar and linguistic mistakes, still major level English language should be thoroughly checked. Please revise your paper accordingly since several language issue occurs on several spots in the paper.
  21. Reference formatting need carefully revision. All must be consistent in one formate. Please follow the journal guidelines.

Author Response

Dear editors and reviewers,

Thank you very much for your decision and constructive suggestions. We have carefully read the comments on our manuscript from the reviewers. And we have revised this manuscript accordingly based on all the points that the reviewers suggested, and some complementary data have been presented in the revised manuscript. Thanks a lot for giving us one more opportunity to address the comments from the reviewers. We sincerely appreciate the reviewers for their constructive suggestions and hope that the revised manuscript is now acceptable.

     You will find a highlighted version of our manuscript, which identify all the modification we made to the manuscript. If there are any other questions or further revision needed, please inform me in your early convenience. All your helps will be greatly appreciated.

Round 2

Reviewer 1 Report

I appreciate the authors for their effort in improving the manuscript to the next level. However, the following minor revisions are required before accepting the paper.

1. Under Section 3.4, still mass 'm' is mentioned as quality in Equation (1). Please correct it to quantity.

2. In abstract, it is still mentioned as '.......considerable application prospects in solar energy systems.' Solar energy systems are the wide spectrum. Mention the specific application of this study.

3. Keywords are not sufficient. Include at least two more keywords.

4. The following relevant articles can be included in introduction section to strengthen the literature review.

(a) https://doi.org/10.3390/en13195079

(b) https://doi.org/10.3390/en13143582

(c) https://doi.org/10.1080/15567036.2019.1607942

Author Response

Dear Editor and Reviewers,

Thank you very much for your review and your suggestive comments. We have corrected the manuscript according to your suggestions.

Reviewer 1#:

  1. Under Section 3.4, still mass 'm' is mentioned as quality in Equation (1). Please correct it to quantity.

Response: Thanks a lot for your careful review and valuable comments. We have corrected quality to quantity in the manuscript.

  1. In abstract, it is still mentioned as '.......considerable application prospects in solar energy systems.' Solar energy systems are the wide spectrum. Mention the specific application of this study.

Response: Thanks a lot for your careful review and valuable comments. We have deleted solar energy systems in the abstract. The specific application of this study has been mentioned as “intelligent temperature management for smart textiles and wearable electronic devices, combining with their solar-thermal energy conversion and storage performance.” in the manuscript.

  1. Keywords are not sufficient. Include at least two more keywords.

Response: Thanks a lot for your careful review and valuable comments. We have added two keywords.

Key words: microcapsules; double-shell; cadmium sulfide; photothermal conversion; heat storage

  1. The following relevant articles can be included in introduction section to strengthen the literature review.

(a) https://doi.org/10.3390/en13195079

(b) https://doi.org/10.3390/en13143582

(c) https://doi.org/10.1080/15567036.2019.1607942

Response: Thanks a lot for your careful review and valuable comments. We have added these three references in the introduction of our manuscript, and the references have been corrected. 

Reviewer 2 Report

Dear Authors
I have reviewed again the manuscript and I think that it is ready for publication. Thank you for considering my suggestions

Author Response

Dear Editor and Reviewers,

      Thank you very much your decision and constructive suggestions.